# Socioeconomic Determinants of Cardiovascular Diseases, Obesity, and Diabetes among Migrants in the United Kingdom: A Systematic Review

**DOI:** 10.3390/ijerph19053070

**Published:** 2022-03-05

**Authors:** Sanda Umar Ismail, Evans Atiah Asamane, Hibbah Araba Osei-Kwasi, Daniel Boateng

**Affiliations:** 1School of Health and Social Wellbeing, University of the West of England, Bristol BS16 1QY, UK; 2Institute of Applied Health, University of Birmingham, Birmingham B15 2SQ, UK; e.a.asamane@bham.ac.uk; 3Department of Geography, The University of Sheffield, Sheffield S1 4DP, UK; h.osei-kwasi@sheffield.ac.uk; 4University Medical Center Utrecht, Utrecht University, 3508 TC Utrecht, The Netherlands; d.boateng-2@umcutrecht.nl; 5School of Public Health, Kwame Nkrumah University of Science and Technology, Kumasi 00000, Ghana

**Keywords:** socioeconomic, migrant, cardiovascular disease, obesity, diabetes

## Abstract

There has been little agreement on the role that socioeconomic factors play in the aetiology of cardiovascular diseases (CVDs), obesity, and diabetes among migrants in the United Kingdom (UK). We systematically reviewed the existing evidence on this association to contribute to filling this gap in the literature. Two reviewers were involved at each stage of the review process to ensure validity. We comprehensively searched through several electronic databases and grey literature sources to identify potentially eligible papers for our review. We extracted data from our finally included studies and appraised the methodological rigour of our studies. A narrative synthesis approach was used to synthesise and interpret the extracted data. We sieved through 2485 records identified from our search and finally obtained 10 studies that met our inclusion criteria. The findings of this review show that there is a trend towards an association between socioeconomic factors and CVDs, diabetes, and obesity among migrants in the UK. However, the picture was more complex when specific socioeconomic variables and migrant subgroups were analysed. The evidence for this association is inconclusive and its causal relationship remains speculative. There is, therefore, the need for further research to understand the exact association between socioeconomic factors and CVD, diabetes, and obesity among migrants in the UK.

## 1. Introduction

Migration to high-income countries has an adverse effect on cardiovascular and metabolic health [1]. Moreover, cardiovascular disease (CVD) and risk factors including hypertension, obesity, and diabetes disproportionately affect migrant populations [2,3,4]. The UK has a growing population of migrants whose states of health pose a threat to public health [5]. Findings from the most recent RODAM (Research on Obesity and Diabetes among African Migrants) study showed a high prevalence of obesity, diabetes [4], and CVDs [6] among Ghanaians residing in the United Kingdom, compared to their compatriots in rural and urban Ghana. South Asians (SAs) are also disproportionately affected by diabetes compared to their home country populations [7].

In the UK, research suggests that the risk of developing diabetes is between two to six times higher in SAs, when compared with white Europeans [8]. Thus, findings from the United Kingdom Asian Diabetes Study (UKADS) showed that SAs tend to have early onset of diabetes and for a longer duration [9]. Among SA subgroups, compared with Europeans, Bangladeshis had the highest odds of developing diabetes, followed by Pakistani and Indians, in a systematic review that assessed the variations in type 2 diabetes (T2D) risk among ethnic minority populations of different geographical origin compared with their host European populations [10]. In addition to the higher burden of T2D among migrant populations, the ways in which they manage this condition are poor compared to Western European populations, resulting in a high risk of death and complications [11]. The rates of ischaemic heart disease (myocardial infarction and angina) are also found to be about 30–40% higher amongst SA men than men in the general population [12]. Age-standardised mortality rate from coronary heart disease (CHD) is reported to be 50% higher among SAs than that of the total population of England and Wales [13]. Death and cardiovascular events among people living with T2D are, for instance, almost 50% higher in SAs than in the Western European population [14,15,16].

More recent studies have explored the potential role of diverse factors in explaining the differences in CVD, obesity, and diabetes risk among migrant populations and the ‘host population’. For instance, in the RODAM study, the prevalence of diabetes decreased with an increasing level of education in Ghanaian men and women in Europe, whereas the association between occupational class and the prevalence of diabetes followed a less consistent pattern in men and women in Europe [17]. These findings suggest the important role that socioeconomic factors can play in the aetiology of non-communicable diseases (NCDs) among migrants; however, the nature of this association remains unclear in the UK context [18]. Whilst a few reviews exist that focus on the factors that influence the health of ethnic minorities [19,20], this has not been performed through the lens of migration. Hence, to our best of knowledge, there has been no systematic review that has consolidated findings from previous studies to understand the socioeconomic risk factors of NCDs among migrants in the UK. The aim of this review was therefore to synthesise, critically analyse, and appraise the quality of the evidence on the socioeconomic determinants of CVDs, obesity, and diabetes amongst migrants in the UK.

## 2. Materials and Methods

This review question is, ‘What are the socioeconomic determinants of CVDs, obesity, and diabetes among migrants in the United Kingdom?’ The reporting of this systematic review followed the Preferred Reporting Items for Systematic Reviews and Meta-Analyses (PRISMA) guidelines [21]. Our review has been registered in the international prospective register of systematic reviews (PROSPERO) with registration number: CRD42021249066.

### 2.1. Eligibility Criteria

Studies were considered for inclusion if they focused on migrants living in the UK (England, Northern Ireland, Scotland or Wales). We defined a migrant as, “any person who is moving or has moved across an international border or within a State away from his/her habitual place of residence” [22]. We were interested in studies that examined the following socioeconomic variables as exposures: income and social protection, education, unemployment and job insecurity, working life conditions, food insecurity, housing, basic amenities, the physical environment, social inclusion and non-discrimination, structural conflict and access to health services. Our outcomes of interest were CVDs, obesity, and diabetes, and we did not limit this to any specific measurement outcomes, such as incidence, prevalence, or severity of outcome. Studies that used observational (cross-sectional, case-control, cohort) or experimental (randomised and non-randomised controlled trials) study designs and published in English language were eligible to be included in our review. There was no limit to the year of publication of studies.

We excluded studies that focused on migrants outside the UK and those that employed case studies or other qualitative study designs. Studies that examined impacts of CVDs, obesity, or diabetes on other outcomes/factors were also excluded.

### 2.2. Information Sources

We searched through the following academic databases for relevant studies: Scopus (1970 to 2021), Medical Literature Analysis and Retrieval System Online (MEDLINE) via Ovid (1946 to 2021), Cumulative Index to Nursing and Allied Health Literature (CINAHL) (1937 to 2021), Embase (1974 to 2021), Applied Social Sciences Index and Abstracts (ASSIA) (1987 to 2021) and Cochrane Library (1996 to 2021). The Web of Science database was searched for grey literature within a content coverage range from 1975 to 2021. All searches in the databases and grey literature sources were last updated on 3 February 2021. We sorted through the reference list of our final eligible studies for other papers.

### 2.3. Search Strategy

We combined several keywords pertaining to our population, exposure, outcome, and settings to search for relevant papers to address our review question. We used truncations and wildcards recognisable by the specific databases to increase the sensitivity of our search. We also used Boolean operators to combine our search terms to obtain the specific papers that were relevant to our review. Our search process, therefore, enhanced the widest possible relevant search for our review. Details of the search terms and their combinations in each of the databases can be found in Appendix A.

### 2.4. Selection Process

Four reviewers (S.U.I., E.A.A., H.A.O.-K., D.B.) were involved in selecting studies for inclusion in the review. Two of the reviewers (S.U.I. and E.A.A.) independently identified papers using the search terms in the various databases. One of these two reviewers (E.A.A.) removed the duplicates from the combined studies identified and used the Rayyan software to automatically exclude papers based on the eligibility criteria [23]. The identified studies were then shared among the four reviewers to screen the titles, abstracts, and full texts of the papers. At the study eligibility stage where the full texts of the papers were assessed, each reviewer validated another’s screening process by randomly selecting four of the studies. Any disagreements were resolved through discussion with all reviewers.

### 2.5. Data Collection Process

We developed a bespoke data collection form to extract data from our included studies which was pilot tested and modified before using it to collect information from the papers. Using this form, we collected information that included the participants’ characteristics, type of study design, recruitment method, socioeconomic variables, outcomes, and results of the association between the socioeconomic risk factors and the outcomes of interest. The included studies were distributed among the four reviewers who then extracted data from the studies, but there was no cross validation of the extracted data.

### 2.6. Study Quality Assessment

We assessed the risk of bias in the included studies using the National Institute of Health (NIH) Quality Assessment Tool for Observational Cohort and Cross-sectional Studies [24] This tool was used to appraise the reliability, validity, generalisability, and overall quality of the included studies using 14 criteria. This included a clearly stated research question and objective, clearly specified study population, adequate participation rate, similar subject selection/recruitment, and uniform application of eligibility to all participants, sample size estimation, exposure measurement before outcome, sufficient time frame to detect an association, examination of different levels of exposure, multiple exposure measurement over time, valid outcome assessment, detection bias, loss to follow-up, and adjustment of confounding variables. The tool provided general guidance to determine the overall quality of the studies and to grade their level of quality as good, fair, or poor. HAO-K and DB assessed the quality of the studies together.

### 2.7. Synthesis Method

A meta-analysis was not appropriate for synthesising the data from our included studies due to two main reasons. There was heterogeneity in the measurement of socioeconomic risk factors and outcomes. Thus, various studies used different indicators to measure socioeconomic variables and CVD, obesity and diabetes. For some studies (see details in Section 3.3) socioeconomic risk factors were controlled concurrently with other non-socioeconomic factors, without sufficient data to examine the exact relationship between the socioeconomic variables and the outcomes. We therefore used a narrative synthesis approach in synthesising the data extracted from our included studies. Thus, we textually organised, described, explored, and interpreted the extracted data on the association and moderators/mediators between socioeconomic risk factors of CVDs, obesity, and diabetes. We started by tabulating the characteristics of the included papers and delineating the findings of the relationship between socioeconomic risk factors and cardiovascular disease, obesity, and diabetes. This was followed by clustering the findings of the study according to the outcomes examined. We then used qualitative case descriptions to explore the nature of the findings within each study by examining key characteristics of each study and the association found between the socioeconomic risk factors and the outcomes. Finally, we examined the findings between the various studies for each outcome category by comparing the nature of the association between our exposures and outcomes between our studies and examining any potential differences, similarities, or explanatory variables for the findings. Where the results were difficult to tell clear associations between the socioeconomic variables and the outcomes, we presented these to show such inconclusiveness on the association.

## 3. Results

### 3.1. Study Selection

Figure 1 illustrates the outcome of the search process. We identified 2485 records from searching through several literature sources. Eleven percent of these papers were identified as duplicates and removed. We then screened the titles and abstracts of the remaining 2266 papers for relevant papers. Ninety-six percent of these papers were excluded at this stage using the Rayyan software [23] because they were not relevant to the focus of our review. The full text of 102 papers were screened and 10 studies were finally included in the review, having met all the inclusion criteria.

### 3.2. Characteristics of the Included Studies

Six out of the ten included studies included migrant populations from a single ethnicity as their sample population [4,6,25,26,27,28]. Among these studies, one study had foreign-born Caribbean migrants [25] and the rest had foreign-born Ghanaian migrants.

For those studies with participants of different ethnicities, two studies involved White, Black, and Asian migrants as participants [29,30]; one study recruited Blacks, Asians, Irish, and White participants [31] and one other study had Indians, Pakistanis, and Bangladeshis as the study population [32]. Table 1 and Table 2 summarise the characteristics of the studies included in the review.

#### 3.2.1. Migrant Population

All included studies had participants aged 18 years and above, except one [31], which had participants aged 16 years and above. All studies included males and females in their sample, except for two studies that recruited only female participants [29,30] (Table 1).

#### 3.2.2. Socioeconomic Risk Factors

The included studies assessed three main socioeconomic risk factors—education, English language proficiency, and socioeconomic status (Table 2). The majority (*n* = 8) of the studies included education as a socioeconomic risk factor [4,6,26,27,28,29,30,32] and two studies assessed English language proficiency as an exposure [31,32]. Three of the studies combined several indices to measure socioeconomic status as a risk factor [6,25,31]. These indices were employment, income distribution, education, car ownership, housing tenure, overcrowding, occupational social class, and area level deprivation (Table 2).

#### 3.2.3. Study Design and Types of Outcomes

The included studies used three types of study designs in their investigations. Five of the included studies used a cross-sectional study design [4,6,26,27,28]; two used longitudinal (cohort) study design [25,29] and the rest analysed existing secondary data that involved national datasets [30,31,32].

Apart from two studies [4,32] which presented results on the association between a range of socioeconomic risk factors and more than one outcome, the rest of the studies reported results on a single outcome. Mainous and colleagues [32] focused on cardiovascular health and diabetes, while Agyemang and colleagues [4] examined obesity and diabetes as outcomes.

Five studies reported CVD as the main study outcome [6,25,27,28,32]. Cardiovascular disease was assessed as either general cardiovascular health, mortality from cardiovascular diseases, undetected elevated blood pressure, 10-year CVD risk estimated from the Pooled Cohort Equations (PCE), or prevalence, awareness and control of hypertension.

For the four studies that reported obesity as the study outcome [4,29,30,31], obesity was measured as child overweight, body mass index, or abdominal obesity. Diabetes was measured by prevalence, awareness, treatment or control of Type 2 Diabetes Mellitus (T2DM) [4,26,32].

All the studies were of good quality, except for one [25] which was judged to be of fair quality. This was because of high loss at follow-up after baseline (>20%). Four of the included studies did not give information on sample size justification, power description, or variance and effect estimates [25,29,30,31].

### 3.3. Relationship between Socioeconomic Risk Factors and Cardiovascular Diseases, Obesity, and Diabetes

Table 2 summarises the results of the association between socioeconomic risk factors and cardiovascular disease, diabetes, and obesity.

Five studies reported results on the association between socioeconomic position, education, English language skills, employment, source of income, and CVD [6,25,27,28,32]. For three of these studies, the association between the socioeconomic exposures and CVD could not be discerned from the results because of the way the exposures were handled in the analyses [25,27,28]. Harding [25] controlled for age and socioeconomic position at the same time when examining the relationship between various risk factors and mortality from CVDs. Agyemang and colleagues [27] adjusted for age, education, and BMI, simultaneously in their investigation of the prevalence, awareness and control of hypertension among various cities including London. There were also no crude results to compare the adjusted results within this study. In van Nieuwenhuizen and colleagues’ [28] study, they adjusted for age, gender, and education level concomitantly when looking at the differences in risks of cardiovascular health between indigenous Ghanaians and Ghanaians living in London.

For the remaining two studies that assessed CVD as an outcome [6,32], their results showed contrasting relationships between socioeconomic factors and cardiovascular diseases. Thus, Mainous et al. [32] reported that greater English language skill was significantly associated with a lower prevalence of undetected elevated blood pressure and previously diagnosed hypertension. These significant associations only existed among Indian, Pakistani, and Bangladeshi migrants for elevated blood pressure and among Indian and Pakistani migrants for previously diagnosed hypertension. Boateng et al. [6] reported that the influence of education, employment, and sources of income combined did not significantly alter the 10-year CVD risk estimate among Ghanaian migrants living in London.

Bijlholt et al.; Agyemang et al. and Mainous et al. reported diabetes as an outcome [4,26,32]. Both Bijlholt et al. [26] and Agyemang et al. [4] examined education as a socioeconomic risk factor, but the influence of this risk factor on diabetes could not be ascertained. This is because this risk factor was controlled together with other sociodemographic variables (Table 2). However, this was not the case with Mainous et al. [32]; the authors reported that greater English language skills were significantly associated with a lower prevalence of previously diagnosed diabetes and a lower prevalence of undetected elevated blood glucose. Nevertheless, this association was only statistically significant for Indian and Bangladeshi ethnicities.

Four studies assessed the relationship between education, income, socioeconomic status, and several indicators of obesity [4,29,30,31]. Two of these studies controlled for their socioeconomic risk factors simultaneously with other exposures; hence, not making it possible to identify the true relationship between the socioeconomic exposures and obesity. Thus, for one of these studies [30], socioeconomic position was concurrently adjusted with other demographic variables (Table 2). In another [4], age and education were simultaneously adjusted for. In these studies, other non-socioeconomic variables would have interacted with the relationship between the socioeconomic risk factors and obesity outcomes. Hence, the nature of the association between these socioeconomic risk factors and obesity was unclear.

The two remaining studies examining obesity as an outcome [29,31], reported a significant association between various social determinants of health and obesity. Non-white native and foreign-born immigrant mothers’ low socio-economic status measured by family income and mother’s education was associated with a lower risk of overweight among children. However, among white immigrant mothers, low income and low education were associated with an increase in the risk of childhood overweight [29]. Thus, the nature of the association between socioeconomic status and childhood overweight differed by the type of immigrant mother.

Socioeconomic status assessed by self-reported occupation, educational qualification and equivalised household income quintiles was significantly associated with waist circumference. However, the nature of the association varied depending on the sex and ethnicity of the migrant. For migrant women, lower socioeconomic status was associated with lower waist circumference among Pakistani, Bangladeshi, Black Caribbean, and Black African ethnic groups. This direct relationship between socioeconomic position and waist circumferences were also observed for Black Caribbean and Bangladeshi migrant men. Higher socioeconomic status was associated with an increase in waist circumference for migrant men who identified as Indian, Chinese, or Black African. However, among Pakistani migrant men, the low socioeconomic position was inversely related to waist circumference. Area deprivation was significantly associated with waist circumference for migrant men and women. For both sexes, as area deprivation increases, waist circumference increases, albeit the association is stronger for men [31].

## 4. Discussion

This is the first review that has synthesised the evidence on the association between socioeconomic determinants of CVDs, diabetes, and obesity among migrant populations in the UK. The findings of this review show that there is a trend towards an association between socioeconomic factors and cardiovascular diseases, diabetes, and obesity among migrants in the UK. However, these findings showed inconsistent patterns in the association between socioeconomic risk factors and the various outcomes.

We found evidence for an association between greater English language skills and low risk of CVDs and diabetes [32], and we judged this evidence to be of good quality. This finding corroborates the results of another study that examined the effect of English language proficiency on coronary heart disease and diabetes within patients in the UK. Among patients registered at practices in London, the odds of preferring non-English language for communication as an indicator of low English language proficiency was 18%, 33%, and 8% more for risk of coronary heart disease, diabetes, and obesity, respectively, compared to preferring English language as the means of communication [33]. Although this study did not intentionally focus on migrants, the majority of the non-English preference participants were migrants in the UK. Higher English language proficiency may determine access to better health care services and information among migrants, which could consequently lead to exposure to healthy lifestyles and protective factors against adverse health conditions [34].

Another important finding from our review about English language proficiency as a socioeconomic risk factor to cardiovascular diseases and diabetes was that, although English language skill might be associated with cardiometabolic and diabetes outcomes among migrant populations in the UK, this relationship seems to be dependent on the ethnicity of the migrants. Thus, it was only among some SA ethnic groups who had a lower prevalence of undetected elevated blood pressure (Indians, Pakistanis, and Bangladeshis) and previously diagnosed hypertension (Indians and Pakistanis) based on their high English language skill. This is, however, contrary to the findings of Mackay, Ashworth, and White where ethnicity did not moderate the association between English language proficiency and cardiovascular diseases and diabetes [33]. We recommend future research to examine the role that ethnicity plays in the association between English language skill and CVD and diabetes to clarify this discrepancy.

We found evidence from two of our included studies for an association between socioeconomic position and obesity among migrants, and both studies were assessed to be of good quality [29,31]. Both studies measured socioeconomic position using different indicators and focused on different ethnic groups (Table 2). One of these two studies focused on childhood overweight and the other focused on adult waist circumference. We found that the relationship between childhood obesity and socioeconomic position varied by the type of migrant—whether non-white, white, or foreign-born. As previous reviews have postulated, the mechanism that explains the way that socioeconomic status leads to obesity is quite unclear in developed countries [35,36], and this is an area where future research is required.

The role that deprivation plays in the risk of obesity can be explained by the limited access to healthy foods and the adoption of unhealthy eating habits and lifestyles, such as the lack of appropriate resources for physical activities in most deprived areas compared to the least deprived areas [37]. There is evidence to show that diet plays a significant role in the association between socioeconomic status and CVD; the quality of diet varies across the socioeconomic spectrum, and the most deprived social groups face a greater burden in terms of risk to CVDs [38]. It is therefore not surprising that we found evidence of a directly proportional relationship between area deprivation and obesity from one of the studies included in our review [31]. Income is shown to be a significant determinant of dietary acculturation [39], and because unhealthy foods (high in saturated fats and sugars) tend to be cheaper than healthy foods [40], groups living in deprived areas, of which migrant populations usually form a significant proportion, may be susceptible to consuming unhealthy foods.

One unanticipated finding in our review was that we did not find evidence for a statistically significant association between socioeconomic status and CVDs among migrants in the UK [6]. This is contrary to expert opinions that conclude that there is a strong link between socioeconomic status and CVDs [41]. The reason for this disagreement is not clear but it may have something to do with methodological differences in the investigation of such relationship, which we have expanded on later in this discussion.

In our review, we found that there were variations in the associations between several socioeconomic risk factors and CVDs, diabetes, and obesity among different ethnic groups. Even among broadly classified ethnicities such as SAs, we found that the risk of CVDs, obesity, and diabetes sometimes varied by nuances in this ethnicity classification, where the nature of the associations was sometimes different for Pakistanis, Bangladeshis and Indians. This is likely to be explained by the socioeconomic as well as genetic and cultural dispositions of the country of origin of different migrants [33]. Thus, socioeconomic factors and some other biological and behavioural factors that are strongly related to CVDs, diabetes, and obesity may be peculiar to particular ethnic groups in the UK. For example, Zaman and Bhopal argue that the higher incidence of coronary diseases among SAs in the UK may be due to the disproportionate distribution of risk factors, such as high smoking prevalence among Bangladeshi men and higher social deprivation among the Bangladeshi ethnic group [42].

It is imperative to understand that aggregating ethnicities into broad labels in delineating the patterns of CVDs, diabetes, and obesity by socioeconomic risk factors does not help conceptualise the social determinants of health that contribute to these health conditions among migrants [43]. An expert review further confirms that the type of migrant group may determine the burden of NCDs within such populations [1].

Contrary to existing evidence consistently showing an inverse relationship between education and cardiovascular risk and diabetes among the general population [44,45], we did not corroborate this evidence from the findings of our review for migrants in the UK. For instance, in a meta-analysis conducted in 2017, comparison between groups of low and medium education versus high education showed an education gradient in cardiovascular risk [45]. Among Ghanaian migrants in Europe, findings from the RODAM study showed a decrease in the prevalence of diabetes with increasing levels of education in Ghanaian men and women. The association between occupational class and the prevalence of diabetes, however, followed a less consistent pattern in both men and women [46].

We believe that, aside from the cultural and genetic differences inherent among the different ethnic groups which could potentially influence the nature, direction and strength of the associations we found in our review, there may be methodological factors explaining this, as well as some disparity in findings between our review and that of previous studies. The conceptualisation of socioeconomic risk factors potentially associated with migrant health in relation to CVDs, diabetes, and obesity is broad. There was substantial heterogeneity in the types of socioeconomic risk factors within our included studies, but also in the wider literature. Socioeconomic status is operationalised in a variety of ways, most commonly as education, social class, or income. The relevance of these measures, and other indicators such as language, acculturation, and integration in assessing the health of migrants needs to be further studied. Moreover, the indicators used to measure socioeconomic risk factors are variable. We also observed significant heterogeneity in the way that the outcomes were assessed in our review and the wider literature in this subject area.

For 60% of the studies included in the review, we were not able to identify the relationship between various socioeconomic risk factors and outcomes due to the way that the data were handled in the analyses of these studies. Thus, in these cases, socioeconomic risk factors were adjusted as confounding variables simultaneously with other variables on the relationships between other exposures and our outcomes of interest. Moreover, results were not presented to depict the exact influence of the socioeconomic risk factors on the outcomes. This, in a way, indicates the minimum focus on the empirical investigation of the social determinants of CVDs, diabetes, and obesity among migrants in the UK.

We employed a rigorous approach at each stage of the review; thus, at least two reviewers were involved at major stages of the review to ensure validity of our approach. Although we limited our review to only English language published studies, we do not believe that this posed a significant limitation on the scope of relevant studies identified, as our focus was on an English-speaking country. Although the evidence that we have presented from this review is observational in nature—that is, all the included studies used cross-sectional or cohort study designs—this is typical of systematic reviews of aetiology and risk. The findings of this review are important for health policy and practice in reducing inequalities in NCDs in the UK, as free hospital access to treatment for NCDs is limited to only legal residents [47].

## 5. Conclusions

Our review highlights a trend towards an association between socioeconomic risk factors and CVDs, diabetes, and obesity in migrants in the UK. However, the picture is more complex when specific socioeconomic variables and migrant subgroups are considered. There was significant heterogeneity in the way the socioeconomic risk factors and cardiovascular diseases, diabetes, and obesity were measured. Our review, therefore, underscores the need to ensure a more consistent conceptualisation and measurement of socioeconomic status among migrant populations to support appropriate recommendations for improving cardiovascular health, diabetes, and obesity in the UK. For example, future studies can adapt measures used in larger UK population surveys to aid in meaningful comparison of their findings. Moreover, the use of socioeconomic status scores and indices, which combine various indicators as well as assess household assets, could be explored in migrant populations in the UK to assess health inequalities.

We were not able to identify the effect of socioeconomic variables on our outcomes from most of our included studies due to the way that the data were handled and reported. This makes the evidence on the association between the socioeconomic risk factors and the outcomes that we examined inconclusive, and causal relationships remain speculative. There is, therefore, the need for further research to consider the association between socioeconomic factors and CVD, obesity, and diabetes as the main research objective and provide sufficient data to understand the exact association between these exposures and outcomes among migrants in the UK.

Moreover, there is also a need for more longitudinal studies that can accurately assess the pathways and impact of the social determinants of CVDs, diabetes, and obesity among migrant populations in the UK.

## Figures and Tables

**Figure 1 ijerph-19-03070-f001:**
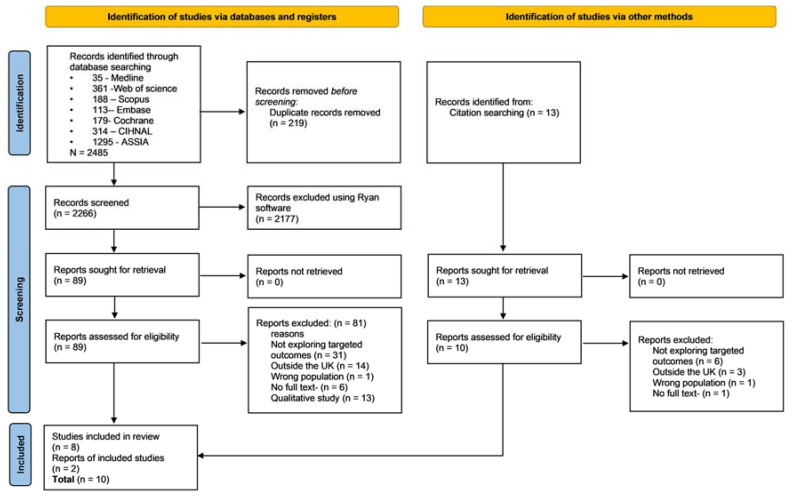
Flow of information through the various stages of the search process.

**Table 1 ijerph-19-03070-t001:** Summary characteristics of the included studies.

Author (Year)	Migrant Population/UK Region	Study Design	Sample Recruitment	Sample Size Included in Analysis	Analysis Method	Socio-Demographic Information	Study Quality
Harding (2004) [25]	Population: Migrants born in the Caribbean Commonwealth countries and aged 25–54 years. UK region: England and Wales	Longitudinal (cohort) study	Analysis of the ONS longitudinal study	1540	Cox regression models	Ethnicity: Caribbean. Age: 25–54 years. Mean year of arrival = 1961.	Fair
Mainous et al. (2006) [32]	Population: Foreign-born South Asian and adults 18 years of age and above. UK Region: England	Secondary data analysis (cross-sectional)	National representative data through random sampling	2523	Chi-square analysis and logistic regression	Ethnicity: Indian, Pakistani, and Bangladeshi. Age: 18–65+ years. Sex: Indian, 49.2% male; Pakistani: 51.5% male; Bangladeshi: 49.3% male.	Good
Martinson, McLanahan and Brooks-Gunn (2012) [29]	Population: White, Black and Asian migrant mothers and children. UK Region: England	Longitudinal (cohort) study	Analysis of the UK’s MCS data—a nationally representative sample of 18,818 children born in the UK between 2000 and 2002	6816	Multivariate Logistic Regression	Ethnicity: White, Blacks and Asians. Mean age at birth: Blacks = 31.3 years and Asians = 28.4 years. Sex: Female (100%). Arrival age ≤ 17 years: Blacks = 20.3% and Asians 44.1%. Arrival age ≥ 18 years: Blacks = 79.7% and Asians 55.9%.	Good
Martinson, McLanahan and BrooksGunn (2015) [30]	Population: Children of migrant and native-born mothers. UK Region: England	Secondary data analysis of national birth cohort data	This study relies on national birth cohort studies that follow children from birth to middle childhood: The MCS	6700	Growth curve modelling-regression	Ethnicity: White, Asian (Pakistani, Bangladeshi, and Indian), and black (Caribbean and African). Mean age: Asian mothers = 27.9 years and blacks = 30.5 years. Sex: Females (100%).	Good
Agyemang et al. (2016) [4]	Population: Ghanaian by country of birth and living in London. UK Region: England	Cross-sectional	Analysis of subset of the RODAM study	1080	Multivariate logistic regression	Ethnicity: Ghanaian. Mean age: men-46.1 years (45.0–47.1) and women-47.7 years (46.9–48.5). Sex: 37.9% men. Mean length of stay in London = 23.2 years. Years since diabetes diagnosis = 10.9 years.	Good
Boateng et al. (2017) [6]	Population: Ghanaian either born in Ghana and either one or both parents born in Ghana (in case of migrants, first generation) or if not born in Ghana, have both parents born in Ghana (in case of migrants, second generation); aged 40 to 70 years without history of CVD. UK Region: England	Cross-sectional	Analysis of subset of the RODAM study. In London, participants were invited based on their registration in Ghanaian organisations	774	χ2 test, ANOVA, and Kruskal–Wallis tests, Density curves, Logistic regression	Ethnicity: Ghanaian. Age: mean = 52 years. Sex: n (male) = 275; n (female) = 499.	Good
Agyemang et al. (2018) [27]	Population: Ghanaian by country of birth and living in London. UK Region: England	Cross-sectional	Analysis of subset of the RODAM study	1080	Multivariate logistic regression	Ethnicity: Ghanaian. Mean age: men 46.1 years and women 47.7 years. Sex: 37.9% men.	Good
Bijlholt et al. (2018) [26]	Population: Ghanaian by country of birth, living in London, aged 25–70 years and having type 2 mellitus (T2DM). UK Region: England	Cross-sectional	Analysis of subset of the RODAM study	632	Multivariate logistic regression	Ethnicity: Ghanaian. Mean age: 54.6 years. Sex: 41.2% male. Mean length of stay in London = 23.2 years. Average years since diabetes diagnosis = 10.9.	Good
van Nieuwenhuizen et al. (2018) [28]	Population: Ghanaian by country of birth and living in London. UK Region: England	Cross-sectional	Ghanaian migrants residing in London were selected from a compiled list of individuals gleaned from population registries or Ghanaian community organisations	3510	Binomial logistic regression and Kruskal–Wallis test	Ethnicity: Ghanaians. Mean age in years (SD): Females 47 (11), Males 46 (12). Sex: 63% Female.	Good
Higgins, Nazroo and Brown (2019) [31]	Population: UK born; child migrant; adult migrant—lived in UK < 5 years; adult migrant—lived in UK 5–9 years; adult migrant—lived in UK 10–19 years; adult migrant—lived in UK 20 years or more. UK region: England	Analysis of secondary cross-sectional data: the Health Survey for England (HSE) (1998, 1999, 2003 and 2004) and the 2001 Census data	HSE provides a nationally representative sample of the population living in private households in England via a multi-stage, stratified, probability sample. Data from the 2001 Census on the area where the HSE participants lived were linked to the HSE data	Model 1 (14,222); Model 2 (14,011); Model 3 (13,673); Model 4 (13,982); Model 5 (13,645)	Multi-level modelling	Ethnicity: Black Caribbean (n = 1331); Black African (n = 376); Indian (n = 1550); Pakistani (n = 1204); Bangladeshi (n = 874); Chinese (n = 804); Irish (n = 1546); and White (n = 20,261). Age: 16–74 years.	Good

RODAM—Research on Obesity and type 2 Diabetes among African Migrants; SES—socioeconomic status; MCS—Millennium Cohort Study; CVH—cardiovascular health; CVD—cardiovascular disease; ONS—Office for National Statistics; CHD—coronary heart disease.

**Table 2 ijerph-19-03070-t002:** Association between socioeconomic determinants of health and cardiovascular disease, diabetes, and obesity.

Author (Year)	Socioeconomic Determinants of Health	Outcomes	Results	How Socioeconomic Determinants Were Handled	Strength of Association between SE Determinant and Outcome
Harding (2004) [25]	Socioeconomic position measured by multiple indices: access to cars, housing tenure, overcrowding, and occupational social class.	Mortality from cardiovascular diseases	After controlling for age and socioeconomic position, the hazard ratios (HR) were imprecise, and the only noteworthy findings were for the oldest age group. Age at migration and duration of residence were independently associated with more than 20% change in circulatory mortality among ages 45–54 years in 1971. A weak positive relationship was also seen for CHD mortality in the oldest age (45–54) cohort.	Adjusted	Unclear because SE determinant was adjusted concurrently with age
Mainous et al. (2006) [32]	Education: assessed as having reported an achieved qualification or not. Self-assessed spoken English language: measured as “very well”, “fairly well”, “slightly”, or “not at all.”	Undetected elevated blood pressure Previously diagnosed hypertension Previously diagnosed diabetes Undetected elevated blood glucose	Greater English language skills were significantly associated with lower prevalence of previously diagnosed hypertension among Indians, Pakistanis, and Bangladeshis. Greater English language skills were significantly associated with lower prevalence of previously diagnosed hypertension among only Indians and Pakistanis. Greater English language skills were significantly associated with lower prevalence of previously diagnosed diabetes among Indians only. It is only among the Bangladeshi ethnic group where a significant association was seen between greater language skills and lower prevalence of undetected elevated blood glucose.	Direct comparison	Significant association
Martinson, McLanahan and Brooks-Gunn (2012) [29]	Education: measured as high and low education. Income: measured as poor (family being in the bottom 30 percent of the income distribution). SES: family income and mother’s education.	Child overweight	Low socioeconomic status is associated with lower risk of child overweight among children of non-white native and foreign-born mothers. For children born to white immigrant mothers, low income and low education are associated with an increase in the risk of overweight.	Direct comparison	Significant association
Martinson, McLanahan and BrooksGunn (2015) [30]	Mother’s education: measured as ‘high’ (have completed A-levels or the vocational equivalent) and ‘low’ (completed O-levels or less) education.	Child BMI trajectory	Relative to White children aged 3 of native-born mothers, Asian children aged 3 of both native- and foreign-born mothers start out thinner but increase in weight at a faster rate that is statistically significant. Black children of native-born mothers have heavier weights at 3 years compared to black children of foreign-born mothers at age 3; however, children of foreign-born mothers increase in weight at a faster rate. These results did not significantly change after controlling for SES and other demographic variables simultaneously.	Adjusted	Unclear because SE determinant was adjusted concurrently with mother’s age at birth, parity and low birthweight status of child
Agyemang et al. (2016) [4]	Education: measured as none or elementary, lower vocational or lower secondary, intermediate vocational or intermediate/higher secondary, higher vocational or university.	Obesity (BMI ≥ 30 Kg/m^2^) Abdominal obesity Type 2 diabetes	The following results were adjusted for age and education simultaneously. The prevalence ratio (PR) of obesity among Ghanaian men in London was 15 times greater compared to that of Ghanaian men in rural Ghana, 15.04 (95% CI 5.98, 37.84). For women in London, the PR was 6.6 times greater, 6.63 (95% CI 5.04, 8.72). The prevalence ratio (PR) of abdominal obesity among Ghanaian men in London was 10 times greater compared to that of Ghanaian men in rural Ghana, 10.48 (95% CI 4.43, 24.77). For women in London, the PR was 2.6 times greater, 2.56 (95% CI 2.25, 2.91). The prevalence ratio (PR) of type 2 diabetes among Ghanaian men in London was 3 times greater compared to that of Ghanaian men in rural Ghana, 3.06 (95% CI 1.67, 5.6). For women in London, the PR was 1.7 times greater, 1.67 (95% CI 1.09, 2.58).	Adjusted	Unclear because results of crude associations not presented, and the SE determinant was adjusted concurrently with age
Boateng et al. (2017) [6]	Education, employment, source of income—no details on these variables provided.	10-Year CVD risk as estimated from the PCE equations for Black men and women.	An association of migration with CVD risk was observed for Ghanaian women living in London compared with those in rural Ghana (OR = 1.45; 95% CI 1.04–2.01). Adjustment for education, employment, and sources of income simultaneously did not significantly alter the risk estimate. A similar case was found for men.	Adjusted	No change in results
Agyemang et al. (2018) [27]	Education: measured as none or elementary, lower vocational or lower secondary, intermediate vocational or intermediate/higher secondary, higher vocational or university.	Prevalence of hypertension Hypertension awareness Control	The following results were adjusted for age, education, and BMI, simultaneously. Adjusted prevalence ratio of hypertension in London compared to rural Ghana was 1.97 (95% CI: 1.58–2.45) for males and 1.51 (95% CI: 1.28–1.78) for females. Age-standardized hypertension treatment ranged from 44% in London in men, and 56% in London in women. The adjusted odds ratio of Ghanaians living in London compared to Ghanaians in rural Ghana was 2.04 (95% CI: 1.28–3.25) for males and 1.51 (95% CI: 1.16–1.95) for females as The adjusted odds ratio of Ghanaians living in London compared to Ghanaians in rural Ghana was 0.86 (0.49–1.58) for males and 0.84 (0.60–1.17) for females.	Adjusted	Unclear because results of crude associations not presented, and SE determinants were adjusted concurrently with age and BMI
Bijlholt et al. (2018) [26]	Education: none or elementary, lower vocational or lower secondary, intermediate vocational or intermediate/higher secondary, higher vocational or university.	Awareness of Type 2 Diabetes Mellitus (T2DM) Treatment of T2DM Control of T2DM	T2DM awareness was 2.7 times higher among Ghanaian migrants living in London compared to rural Ghanaians, OR = 2.7 (95% CI: 1.3–5.6). Adjustment for age, sex and level of education concurrently did not have any effect on the odds ratio; OR = 2.7 (95% CI: 1.2–6.0). T2DM treatment was 4 times higher among Ghanaians in London compared to rural Ghanaians, OR = 4.0 (95% CI: 1.9–8.3). Adjustment for age, sex and level of education concurrently slightly reduced the odds of treatment between rural Ghanaians and Ghanaian migrants in London to 3.4 (95% CI: 1.5–7.5). Control of T2DM was comparable between rural Ghanaians and Ghanaian migrants in London, OR = 0.4 (95% CI: 0.2–0.9) and this association remained after adjusting for age, sex and level of education simultaneously, OR = 0.4 (95% CI: 0.2–0.9).	Adjusted	Unclear because SE determinant was adjusted concurrently with age and sex
van Nieuwenhuizen et al. (2018) [28]	Education: none or elementary, lower vocational or lower secondary, intermediate vocational or intermediate/higher secondary, higher vocational or university.	Cardiovascular Health	Relative to rural Ghanaians, Ghanaians in London had 95% lower odds of having 6 or more components of ideal cardiovascular health (Crude OR = 0.050 (0.026–0.095; *p* < 0.001). After adjustment for age, gender and education level simultaneously, the odds ratio only reduced slightly, OR = 0.043 (0.021–0.087); *p* < 0.001 with no change in the association.	Adjusted	Unclear because SE determinant was adjusted concurrently with age and sex
Higgins, Nazroo and Brown (2019) [31]	English language proficiency (reads or speaks English). Socio-economic characteristics measured using Registrar General Social Class based on self-reported occupation; highest educational qualification; equivalised household income quintiles; area level deprivation-measured using the Index of Multiple Deprivation 2004 variable.	Obesity (continuous waist circumference)	For women, the addition of socio-economic characteristics results in notable further reductions to the waist circumference of those ethnic groups with the lowest socio-economic status (the Pakistani and Bangladeshi groups, followed by the Black Caribbean and Black African groups), relative to White women. For example, the coefficient for Bangladeshi women reduces from 4.36 cm to 3.22 cm, relative to White women. Similarly for men, the addition of the socio-economic characteristics block of variables results in notable further reductions to the waist circumference of those ethnic groups with lower socio-economic position (Black Caribbean and Bangladeshi men), relative to White men, but also increases the coefficients of those with a higher socio-economic position (Indian, Chinese and Black African men). For Pakistani men (who have a low socio-economic position) the waist circumference coefficient increases, relative to White men, when socio-economic characteristics are added to the model. When area deprivation was included in the socio-economic status block, there was a strong association between area deprivation and waist circumference for both men and women—waist circumference increases as area deprivation increases. The association is particularly strong for Men—for example, men who live in the most derived areas have a waist circumference 0.90 cm greater than those who live in the least deprived areas.	Adjusted	Significant association

CVD—cardiovascular disease; CHD—coronary heart disease; BMI—body mass index; SES—socioeconomic status; OR—odds ratio; PCE—Pooled Cohort Equations.

## Data Availability

Not applicable.

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
