# Peer review of "Socioeconomic Determinants of Cardiovascular Diseases, Obesity, and Diabetes among Migrants in the United Kingdom: A Systematic Review"

_ijerph, 2022, doi:10.3390/ijerph19053070_

Round 1

Reviewer 1 Report

Dear Authors,

thank you for the opportunity to review this very valued work that points out an important topic of the health status of migrants in the UK. First of all, let me just briefly state that I enjoyed reading the paper very much, and as a guideline and evidence synthesis methodologist and public health researcher as well as a person living away from the country of origin, the paper hits home.

The systematic review is conducted using a robust and transparent methodology of the highest current standard. The search strategy is comprehensive and supplemented in full, which is also shown in the number of screened records. The results are presented clearly and the paper has an excellent standard of language and writing.

I have a few minor suggestions to take the paper just one small step further. This is only because I consider it a very important piece of work. 

Introduction

I find the meaning of what you state in the introduction in some places confusing, especially in the connection to the "healthy migrant" concept. Some statements seem contradictory but this could be my lack of understanding. Is the migrant health generally better or poorer when compared to the country of origin and/or host country? "African migrant populations for instance have higher CVD risk than their host and native 34 home country populations"..... "high prevalence of obesity and diabe-36 tes [6], and CVDs [7] among Ghanaians residing in the United Kingdom compared to their 37 compatriots".............. "South Asians (SA) are also disproportionately af-38 fected by diabetes compared to their host and home country populations"

VERSUS "Existing literature has explained these observations using the "healthy migrant" con-56 cept; a concept that refers to the positive selection of international migrants who move in 57 search of better labour market opportunities, because they have the resources and moti-58 vation to move." Which suggests that the migrant health is better..? Followed by another seemingly contradictory statement: "Research among Mexicans residing 60 in the United States also suggests that the Mexican health advantage may be attributable 61 to selective migration among less healthy migrants."

Please, revise if it´s written the way it was intended to.

In the introduction, please add rationale on why the design of systematic review of aetiology and risk and the review objective help answer this issue.

In the introduction, state whether a similar published or ongoing review has been done. (Searches in Epistemonikos, PROSPERO, etc.) This will also support the statement in discussion: This is the first review that has synthesised the evidence on the association between 42 socioeconomic determinants of CVDs, diabetes and obesity among migrant populations 43 in the UK

Methods

At the beginning of the methods section, include an explicit review question using the PEO format, using the following components: Population (types of participants); Exposure of interest (independent variable); Outcome (dependent variable)

and a reference to the PRISMA guidelines.

Include a filled-out PRISMA report in the supplementary files.

State whether a protocol had been published or registered a priori, and if not, include a brief explanation.

Did you use any method to limit the risk of extraction errors? (e.g., two reviewers extracting independently)

Explain, why meta-analysis was not done. (I suppose because of heterogeneity of data. In that case, it would be useful to provide specific recommendations for future research to enable meta-analysis of data from the studies in the future.) This is connected to what you state in the conclusion: "We were also not able to identify the effect of socioeconomic variables on our outcomes from most of our included studies due to the way that the data was handled and reported." (lines 165-166) Can you elaborate? How was data handled and reported and how should it be in future studies?

Include the data extraction form in the supplementary files, or at a minimum, specify the exact items in the methods.

Results

In Figure 1, is the software used Ryan, or rather Rayyan? Any software used for citation screening, extraction, or other steps should be mentioned and cited in the methods.

Line 174 - White, Blacks and Asian migrants (either Whites, Blacks, or White, Black..)

Line 179 - except [32], (include a description of the study, e.g. except for the study by Author [32] or "except one [32]")

Page 8, line 3 - delete "cite"

Conclusion

In conclusion, please add the population and country characteristics in the first sentence (line 156): "Our review highlights a trend towards an association between socioeconomic risk factors and CVDs, diabetes and obesity." ... in migrants in the UK.

Could you suggest specifically how future researchers should measure SES? "more consistent conceptualisation and measurement of socioeconomic status among migrant populations" (line 161).

You state the following both in the discussion and in the conclusion: "The evidence we synthesised from our included studies comes from only observational studies." (line 164) I wonder if it´s possible to carry out RCTs or experimental studies in this topic; I don´t think so. This is not a limitation, or at least not an important one. It´s typical to have observational studies in systematic reviews of aetiology and risk. To briefly state this in the discussion is sufficient, delete from conclusions.

References should not use bullet-points, only a numbered list.

Thank you very much for this rigorous work advocating for the health of migrants in the UK. I am looking forward to another read.

Author Response

Dear Reviewer,

Thank you for your feedback and comments in helping us to consider strengthening various aspects of our article. We found your comments very useful. Please find attached below the ways we have addressed your comments (in red).

Reviewer 2 Report

I would like to sincerely thank the authors for the research carried out on an intertwined subject matter of health and migration. Both are of relevance in terms of research and policy-setting. It was my pleasure to review the draft.

The article offers an informative overview of research and analysis carried out, with a clearly described methodology. I have found it informative and engaging to read.

Certain findings and conclusions that came out clearly, such as the correlation between the knowledge of the EN language and specific types of health issues, contain a direction for policy and educational/ awareness work.

Abstract offers, as it should, a synopsis. A number of minor suggestions for consideration below.

  1. Introduction

30-32:  “Migration to high-income countries generally has an adverse effect on cardiovascular and metabolic health [1]” – the link to the citation does not work. The elements of the article cited available in open access seem to suggest a varying degree of CVD risk amongst countries of origin/destination and duration of stay, which is not quite the same as the opening statement.

33: a suggestion to refrain from using value statements in academic writing, as the words such as “important” introduce a hierarchy.

30-54: this information is relevant to the article, as the facts/findings are explained in the subsequent para.

72-74: a policy/applied angle enhances the relevance of the study.

  1. Materials and Methods: are clearly presented, including the eligibility criteria, sources, and methodology for approaching the cases.
  2. Results - 3.1 seems well suited for the methodology section. Inclusion of Figure 1. Is a plus, as it is visual, illustrative, and logical.
  3. Discussion draws attention to the role the ethnicity plays in the association between English language skill and CVD and diabetes. Valid explanations are provided to the findings. Categorisation of migrants by countries of origin and the impact of customary health habits are well reflected.
  4. Conclusions are solid and offer strands for further research.

Additionally, the draft complies with standards for accuracy in references.

Author Response

Dear Reviewer,

Thank you for your feedback and comments in helping us to consider strengthening various aspects of our article. We found your comments very useful. Please find attached the ways we have addressed your comments (in red).

Reviewer 3 Report

Thank you for the opportunity to review this paper, which is a much-needed addition to an important area of investigation, given the health inequities manifesting in immigrant populations. The methodology is robust; however,  the discussion is at times lacking in succinctness and the flow of the narrative can be improved to convey more clearly the findings of this review.

Some specific recommendations:

Introduction.

  • Line 32: indecorously is not a word often found in scientific papers. It relates to taste/manners, and the sentence would be better if this was changed to another word that conveys inequity.
  • Lines 48-50: “management has been poor compared to the Western European populations resulting in a high risk of deaths and complications” …. Is there a reference for this statement? Otherwise, can this be asserted to be directly associated to poor management, given the higher burden of disease that is stated in the following sentences? It may be due to poor access to health care, poor health literacy, poor adherence, or many other factors other than poor management.
  • Lines 70-71: “socioeconomic factors play an important role in the aetiology of NCDs among migrants” needs reference
  • Line 71: acronym NCDs needs to be defined.
  • Line 75: “The aim of this review is to synthesise evidence on the socioeconomic determinants”. The aims are more than this, as pointed out in the abstract, the paper also critically analyses the evidence, and should be stated. The paper not only presents a synthesis of the evidence it also comments on the quality of previous research and associations made.

Discussion.

  • Lines 44-45: It would be beneficial for the discussion to articulate the major findings of your review in the first paragraph, that you summarised well in the abstract "The findings of this review show that there is a trend towards an association between socioeconomic factors and cardiovascular diseases, diabetes, and obesity among migrants in the UK". Then, this could be qualified by statements about the inconsistencies, and further exploration of the associations or lack thereof.
  • lines 54-55: “the majority of the non- English preference participants were migrants in the UK.” Is this assertion from the study you quoted in the previous sentence?
  • lines 91-91: “migrant populations may be susceptible to consuming unhealthy foods.” It is not clear from the evidence presented in the paper why there is a difference between different ethnic groups living in the areas of deprivation compared to white residents. What evidence supports the above statement, and why would this be different to white people living in the same area of deprivation? There is also good evidence to support that not only is whole/ healthy food more expensive than calorically dense food, but whole food is also more expensive in areas of deprivation compared to well resourced areas. This would apply to all groups living in this area.

Conclusion.

  • In line 169, “This indicates a need for further research”. The thrust of your narrative would be helped by more assertive language than this sentence in your conclusion. 

Author Response

(The authors gave the same response as above.)

Reviewer 4 Report

This is a systematic review of epidemiological studies aiming to synthesize the socioeconomic determinants of cardiovascular diseases, obesity, and diabetes, among migrants in the United Kingdom.

Title: May consider taking out “of Epidemiological studies”.

Abstract: Sections (i.e. background, methods, etc.) are clear though I suggest improving conciseness.

Introduction: The healthy migrant concept (paragraph 4) does not seem to explain the observations mentioned above that paragraph. The statistics stated before point out that migrant health tends to be worse thus these arguments are inconsistent. As well, it is not clear why socioeconomic determinants are the focus of this paper. Consider elaborating on the last point about lack of free hospital access for migrants.

Methods: Outcomes are unclear - are you looking for new diagnosis, worsening of condition, or otherwise? What were acceptable measures of these outcomes?

Results: Colour coding or re-organizing table by meaningful categories based on either the social determinants or the results may improve understanding. Otherwise, section is well done. Concluding sentences of paragraphs are clear and strong.

Discussion: In paragraph 3, I suggest identifying an area for further research rather than saying “It is difficult to explain…” in the last sentence. Similar issue in paragraph 4. Uncertainties should be described as areas for further research or as limitations of this review, rather than a personal doubt phrased with “we are not sure…”. Drawing on existing evidence to speculate hypotheses for future studies would strengthen this section. As well, there seems to be a gap in why socioeconomic factors are important - what are the implications of this study clinically or policy-wise?

Conclusion: From the results, are there certain outcome measures and risk factors that are most consistent across studies? Perhaps those ought to be highlighted as needing further research - otherwise the issue of heterogeneity will persist and this paper’s call for further research does not contribute much to the literature. Apart from that, this section is clear and summarizes the difficulty of this area of study well.

Writing style: Overall, succinct and clear. Few grammar and tense errors throughout.

Author Response

(The authors gave the same response as above.)
